

# Farming by soil in Europe: status and outlook of cropping systems under different pedoclimatic conditions

Gergely Tóth[1,2], Tamás Kismányoky[1], Piroska Kassai[1], Tamás Hermann[1], Oihane Fernandez-Ugalde[3] and Brigitta Szabó[2]

[1] Georgikon Faculty, University of Pannonia, Keszthely, Hungary
[2] Institute for Soil Sciences and Agricultural Chemistry, Centre for Agricultural Research, Budapest, Hungary
[3] Directorate D—Sustainable Resources, European Commission, Joint Research Centre, Ispra, Italy

Corresponding author
Piroska Kassai,
kassai.piroska@georgikon.hu

## ABSTRACT

**Background**. Despite of the importance of soils in agronomy, to date no comprehensive assessment of cropping in Europe has been performed from the viewpoint of the soil variability and its relationship to cropping patterns. In order to fill this knowledge gap, we studied the cropping patterns in different soils of European climate zones with regards to the shares of their crop types in a comparative manner. The study highlights the main features of farming by soil in Europe. Farming by soil in this context means the consideration of soil characteristics when selecting crop types and cropping patterns.

**Methods**. We first assessed the dissimilarity between the cropping compositions of different pedoclimatic zones in Europe. Next, we assessed the differences of crop distribution in the climate zones by soil types and main crop types by analyzing the degree of association of crops to soil types. A detailed country scale assessment was performed using crops-specific soil productivity maps and land use survey data from Hungary.

**Results**. Results suggest that, in general, farmers consciously take pedoclimatic condition of farming into account when selecting their cropping patterns. In other words, farming by soil is a common practice in the different climatic regions of Europe. However, we have strong reasons to believe that soil suitability-based cropping is not practiced to its full potential over the continent. For example, the findings of our European assessment suggest that production areas of legumes are not always optimized for the local pedoclimatic conditions in some zones. These findings also underline that economic drivers are decisive, when farmers adopt their cropping (eg. oil crops on Albeluvsiols in Europe). Win-win situations of economic considerations and soil suitability based management are observed in all pedoclimatic zones of Europe. The country analysis shows that cropping is progressively practiced on more suitable areas, depending also the crop tolerance to variable pedoclimatic conditions In conclusion, we can assume that pedoclimatic conditions of cropping are respected in most of Europe and farmers crops according to edaphic conditions whenever economic considerations do not override the ecological concerns of farming.

## INTRODUCTION

Sustainable agriculture aims to maintain productivity by optimizing the use of locally available resources, including climatic and edaphic resources and reducing the use of external and non-renewable inputs. Improving the synergies between soil qualites and cropping patterns is key to achieve this aim. While the principle of optimization of plant production for local conditions is widely recognized (*Wezel et al., 2014*), there is little knowledge about its practice in Europe. According to the common scientific understanding, the importance of economic and policy factors play a key role in farmers' decision when selecting the most appropriate crop type (*Tilman et al., 2002*). Farmers go for profit, even if sustainability of the production is among their considerations. The Demands of the world commodity market and subsidy schemes of the Common Agricultural Policy influence the profitability of cropping in the European Union and thus the planning of the entire cropping system (*Van Zanten et al., 2014*). While economic and policy factors delimit the farmers' optimization strategies, the characteristics of underlying ecological resources set the boundary conditions for farming. Intensive agricultural management can melt away much of the difference in the suitability for cropping different plants on given site, at least to a degree which turns the ecologically less suitable alternative to an economically more profitable one (*Van Ittersum & Rabbinge, 1997*). Hence, farming by soil can be part of the solution for sustainable intensification, which targets high economic profit with resource use optimization and conservation (*Pretty, 1997*). Farming by soil in this context means the selection of the most suitable cropping system for a given soil. Application of the most appropriate soil management technique is the other main component of soil specific farming and has received more attention with the emergence of precision agriculture in the past decades (*Bongiovanni & Lowenberg-Deboer, 2004*). However, so far little scientific research has been made to assess the relationship between soil quality and composition of crops.

Our analysis aimed to fill this information gap and targeted to reveal the relationship between cropping practices and soil conditions in Europe and with a detailed assessment in Hungary. Soil quality, including water management and nutrient availability is an essential, yet quite variable property of soil types, therefore we performed our analysis for Europe based on soil types, defined as Reference Soil Groups (*FAO, ISSS & ISRIC, 1998*). To enable integrated analysis of complexity in the climate-soil-crop system, cropping pattern by soil types were assessed in the main climate zones. In a country analysis, where (i) crop-specific soil productivity maps, showing the suitability of cropping and (ii) field survey data on major crops were available, we assessed the crop distribution of major crop types on soils with different productivity levels.

Our analysis focused on land-based agriculture, i.e., large scale open-air arable farming and did not assess differences in management practices. While recognizing the influence of land management on the success of farming, our current interest was solely in testing the hypothesis of existing spatial relationship between soils and farming systems in the continent of Europe and on a country scale in Hungary.

## MATERIALS & METHODS

### Identification of cropping system in pedoclimatic zones of Europe using spatial datasets

Spatial units of the analysis were the pedoclimatic zones (PCZs), which are unique combinations of soil type and climate. Pedoclimatic zones were delineated by overlapping the layer of European climatic zones geodatabase, as reclassified for soil productivity evaluation (*Tóth et al., 2013* based on *Hartwich et al., 2005*; see Fig. 1.) with the layers of Reference Soil Groups (RSGs, *FAO, ISSS & ISRIC, 1998*) of the European Soil Database (ESDB, *European Commission, 2003*). Physical area for crops was calculated for each pedocimatic zone using the MapSpam 2005 dataset developed by *You et al. (2014)*. Spatial precision of the MapSpam data was assessed by *Joglekar, Wood-Sichra & Pardey (2019)* and their findings confirm the adequacy for a using this data together with the ESDB data, for continental scale analysis. As a result, we created geodata layers containing the information of the physical area covered by each crop in each pedoclimatic zones of Europe. We focused on the following cropping system classes: cereals (barley, millet pearl, millet small, sorghum, wheat and other cereals), maize, legumes (bean, chickpea, cowpea, lentil, pigeon pea, and other pulses), oil crops (rapeseed, sesame, soybean, sunflower and other crops) and root crops (cassava, potato and other roots and tubers).

### Statistical analysis to reveal spatial characteristics of crop systems in relation to pedoclimatic conditions in Europe

The analysis of crop systems in the pedoclimatic zones of Europe was performed by main climate zones, i.e., the climatic component of the pedoclimatic zones. Similarities and differences of the distribution of crop types on different soils within climatic zones were assessed in a comparative manner using dissimilarity analysis.

We first assessed the dissimilarity between the cropping compositions of different pedoclimatic zones.

Dissimilarity was calculated with 'vegan' package (*Oksanen et al., 2017*). We computed Bray–Curtis dissimilarity index with vegan R package to quantify dissimilarity of RSGs based on the distribution of their cropping systems in pairwise comparisons. The dissimilarity index reaches its maximum value one when there are no shared cropping system classes between two compared RSGs. Analysis has been performed for each climatic zones separately. Function decostand was used to standardize the values.

The matrix of dissimilarity index shows the differences between the compositions of cropping systems of the compared PCZs by climate zone. The darker the cell in the matrix, the larger the difference between the cropping compositions of the two compared PCZs (soil types) within the given climate zone.

Next, we assessed the differences of crop distribution in the climate zone by soil types (RSGs) and main crop types, by analyzing the degree of association of crops to soil types (RSGs).

Association was analyzed with 'vcd' R package (*Meyer, Zeileis & Hornik, 2016*). Association plot (*Meyer, Zeileis & Hornik, 2003*): reject the null hypothesis of independence of the two categorical variables when there are residuals which are too extreme, i.e., not

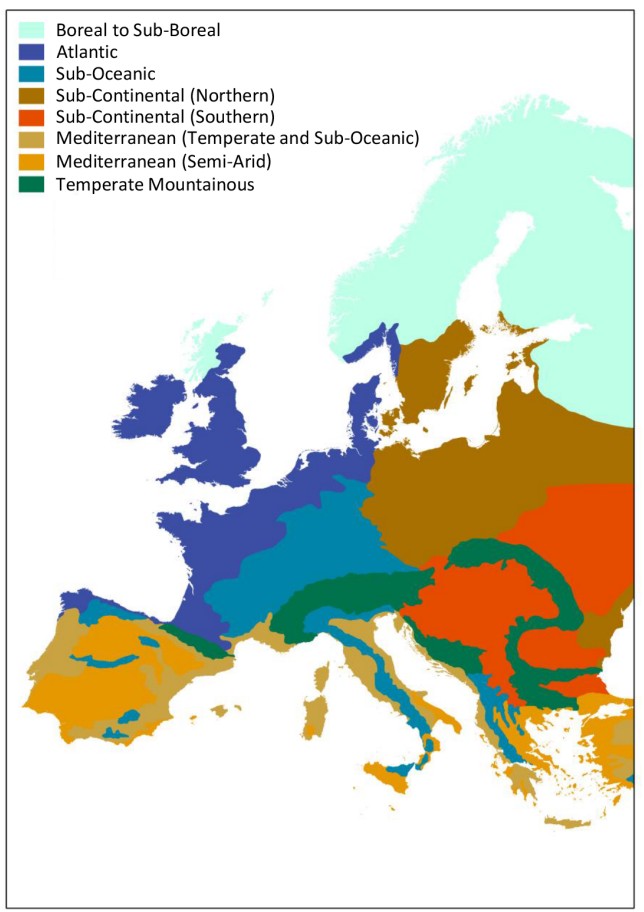

**Figure 1** **Climate zones in Europe.**

close enough to zero. This is the well-known $\chi 2$ test for independence in 2-way tables. When the $\chi 2$ test statistic turns out to be significant for some data, it seems natural to go back to its components, i.e., the residuals, for a more detailed analysis. Association plots visualize the table of Pearson residuals (*Zeileis, Meyer & Hornik, 2007*; *Meyer, Zeileis & Hornik, 2006*): each cell is represented by a rectangle that has (signed) height proportional to the corresponding Pearson residual and width proportional to the square root of the expected counts. The highlighted cells are those with residuals individually significant at approximately the 5% and 0.01% level. The main purpose of the shading is not to visualize significance but the pattern of deviation from independence."

We analyzed with Chi-squared test (*Everitt & Hothorn, 2010*) if RSGs are significantly different regarding the distribution of the farming systems. Test has been performed on each RSG pairs.

Those soil types are considered, which occupy at least 1% of the area of the climate zone and of which at least 10% is cultivated.
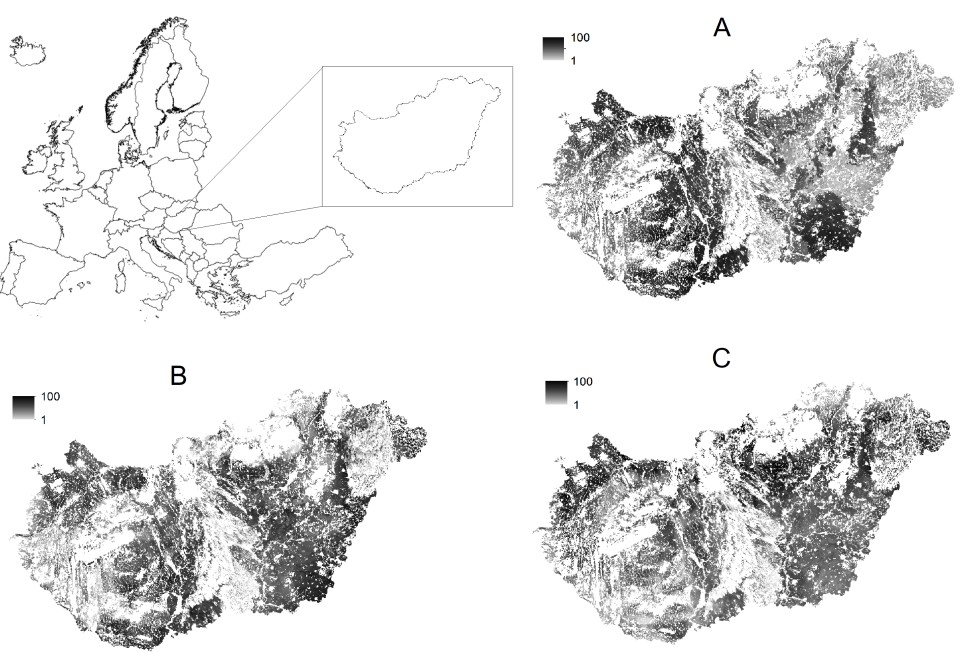

**Figure 2** **Crop-specific soil productivity in agricultural areas of Hungary.** (A) Wheat; (B) maize; (C) sunflower; productivity score: 1, lowest, 100, highest productivity.

## Analysis of crop distribution by cropland soil productivity in Hungary

The favorable conditions of data availability from field level crop-specific soil productivity maps and crop data from ground survey enabled to extend our analysis both regarding spatial and thematic accuracy within one European country, namely in Hungary.

Plant-specific soil productivity maps with 100m spatial resolution were used to characterize the suitability of crop production. Wheat, maize and sunflower productivity maps based on *Tóth (2009)* are available for the whole of the cropland area of the country as composed by *Tóth et al. (2018)*. Cropland areas were classified on a scale 1–100 based on the degree of suitability provided for growing major crops, 1 indicating the least productive and 100 is the most productive land (Fig. 2).

Spatially explicit wheat, maize and sunflower observations collected during the Land Use/Cover Area frame Survey (LUCAS) campaigns in 2009, 2012 and 2015 (*EUROSTAT, 2020*) were used to indicate cropping patterns on croplands in Hungary. 1,247 wheat, 1,495 maize and 940 sunflower crop fields were observed in Hungary during the three campaigns. Data of the crop observations were superimposed on the crop-specific productivity maps.

Share of crop observations by soil productivity classes (low, medium, high, very high, including land with productivity scores 1–40, 40–60, 60–80 and 80–100 respectively) were calculated and number of cases were compared among classes.

All spatial analysis was performed in ArcGIS 10.4.

**Table 1** Differences in farming systems between Reference Soil Groups (RSGs) in different climate Zones of Europe (result of Chi square statistics, $p \leq 0.05$).

| Climate zones | RSGs | other RSGs with significantly different farming systems |
|---|---|---|
| 1. Atlantic | Histosols: | Albeluvisols, Arenosols, Cambisols, Gleysols, Luvisols, Leptosols |
| | Arenosols: | Albeluvisols, Fluvisols, Histosols |
| 2. Sub-oceanic | | There is no significantly difference between any RSG |
| 3. Northern-sub oceanic | Khastanozems: | Significantly different from all other reference soil groups |
| 4. Mediterranean semi-arid | Acrisols: | Significantly different from all other reference soil groups |
| | Calcisols: | Acrisols, Regosols, Vertisols |
| | Fluvisols: | Acrisols, Vertisols |
| 5. Southern sub-continental | Podzols: | Significantly different from all other reference soil groups |
| | Gleysols | Histosols, Leptosols, Pozols |
| 6. Mediterranean (temperate and sub-oceanic) | Calcisols: | Fluvisols, Luvisols, Podzols |
| | Podzols: | Calcisols, Cambisols, Leptosols |
| 7. Temperate mountainous | Gleysols | Significantly different from all other reference soil groups |

## RESULTS

### Crop systems in relation to pedoclimatic conditions of the Atlantic climate zone

According to the statistical comparison (Table 1) displayed also in the evaluation matrix (Fig. 3), cropping pattern of Histosols differs from those on most other soil types except for Fluvisols and Podzols to the greatest extent. Furthermore the cropping pattern of Albeluvisols is significantly different from those of Arensols, while the cropping pattern of Arenosols is also significantly different from that of Fluvisols (besides Albeluvisols and Histosols).

The difference in the pattern of Histosol cultivation is largely due to the relatively high share of root crops on this soil type, which is cultivated on a significantly higher share of the area of this soil than on other soil types in the Atlantic climate zone (Fig. 4). Histolsols are soils of loose structure, which is optimal for growing root crops. Therefore favoring root crops on this soil can be considered as good agronomic practice. According to our findings root crops are given preference over maize, and to some extent oil crops on these soils. Potato and sugar beet are predominant among root crops of the climate zone. Both crops are selective in their fore crops and may return to the rotation after 4 years. The high spatial extent of root crops on Histosols suggest that farmers aim the maximum capacity of root crop (potato and sugar beet) production on these soils, considering agronomic possibilities.

On Albeluvisols areas proportionally more oil corps (rapeseed) and maize are grown than on Arenoslos and Histosols. In the meantime the proportional areas of root crops and cereals are smaller. As Albeluvisols are generally low fertility soils not particularly suitable for arable cropping, the relatively higher share of cash crops may be a result of intensive cultivation driven by economic incentives, rather than a search for the most suitable crops.

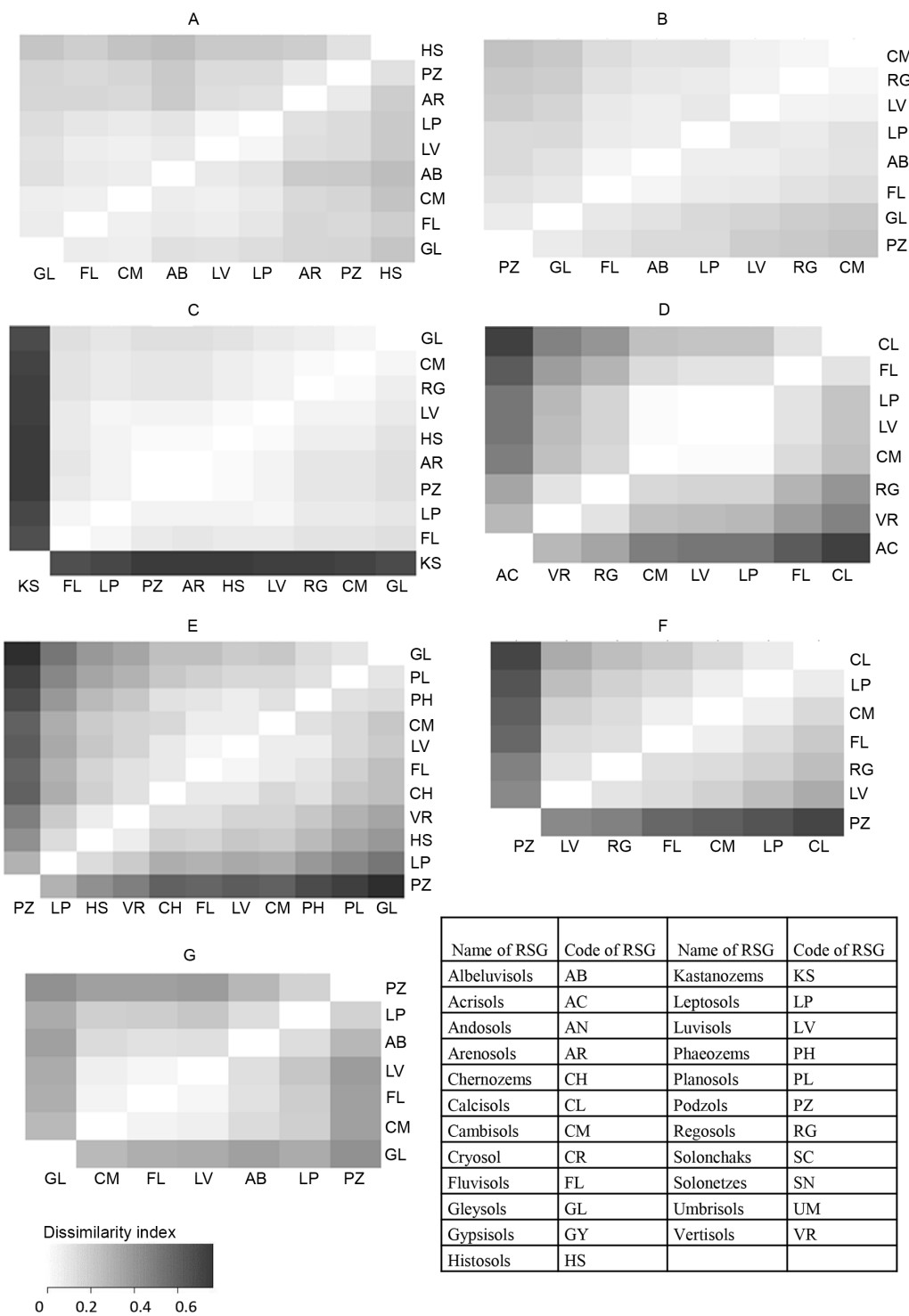

**Figure 3** **Dissimilarity indices matrix of cropping systems by Reference Soil Groups.** (A) Atlantic climatic zone (B) Sub-Oceanic climate zone (C) Northern sub-continental climate zone (D) Mediterranean, semi-arid climate zone (E) Southern Sub-Continental climate zone (F) Mediterranean (temperate and sub-oceanic) climate zone (G) Temperate mountainous climate zone.

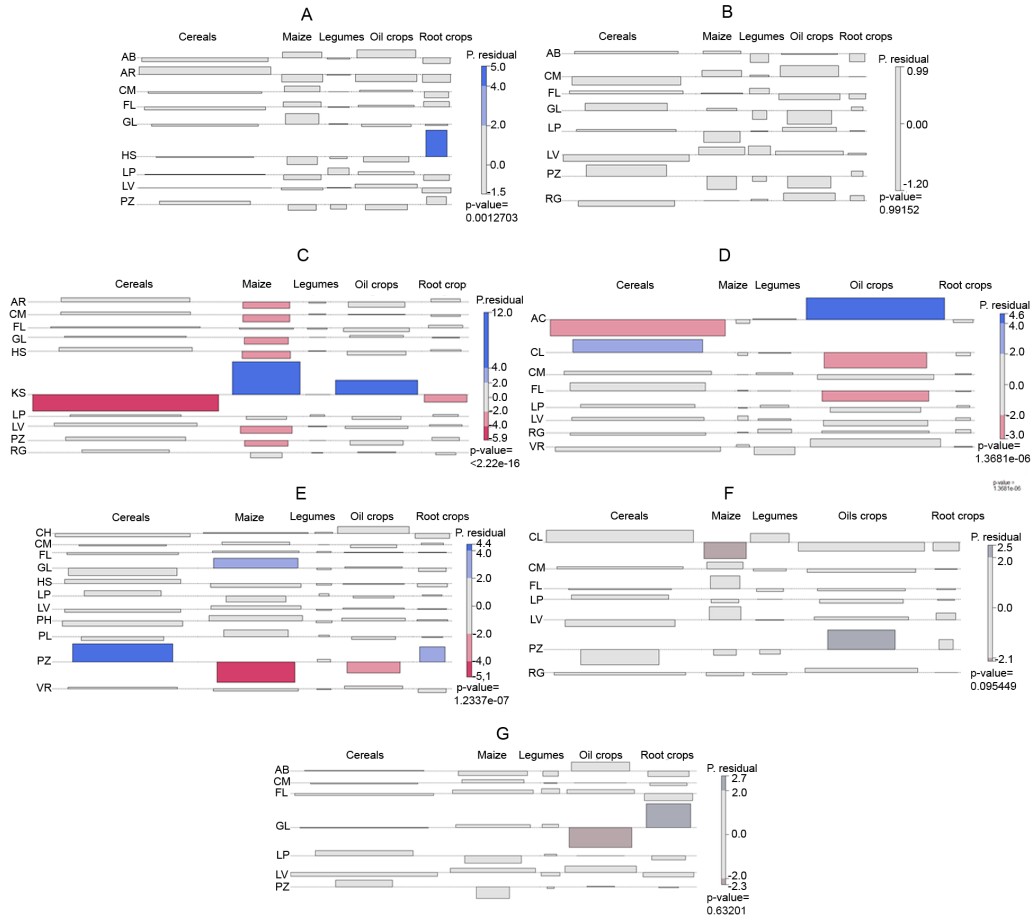

**Figure 4 Conditional association plots of cropping systems by Reference Soil Groups.** (A) Atlantic climatic zone (B) Sub-Oceanic climate zone (C) Northern sub-continental climate zone (D) Mediterranean, semi-arid climate zone (E) Southern Sub-Continental climate zone (F) Mediterranean (temperate and sub-oceanic) climate zone (G) Temperate mountainous climate zone.

Dominancy of cereals on Arenosols can be explained by the advanced technologies of high intensity farming for these crops. Moisture availably, which is often the limiting factor for cereal cultivation is secured under the Atlantic climate. Easy workability and weed control along with well-planned nutrient management can secure high returns from cereals in this pedoclimatic zone.

## Crop systems in relation to pedoclimatic conditions of the Sub-Oceanic climate zone

Cropping on Podzols and Gleysols is somewhat different from those on the rest of the soil types (Figs. 3 and 4). A higher share of cereals and lower share of oil crops is observed in the cultivation pattern of these two soils in the Sub-Oceanic climate zone. However, these differences are not significant. Nevertheless, cultivation of cereals on Podzols is successful if moisture is available and fertilization is adequate. Both conditions are given in this climate zone and its countries with agriculture of advanced technological levels.

Rapeseed, which is the predominant oilcrop in the region and which is primarily cultivated for animal feed and also for canola oil, has a need for quality seedbed preparation and high input of nutrients. The easy workability of Podzols would make them optimal for rapeseed production. However, disparity between the high nutrient demand of rapeseed and the low nutrient supply of Podzols can be lessened probably only with such high inputs, which are not economic. Cambisols and to some extent Luvisols, Leptosols and Regosols of the region are more suitable for rapeseed than for more demanding crops like maize, which is reflected in the cultivation share of those soils.

Cereals and oil crops have similar share on Gleysols to those on Podzols. However, Gleysols have higher share of maize. Gleysols are hydromorphic soils, of which the subsoil is wet for most part of the year, therefore cultivating maize—a crop of high water and nutrient demand—can be regarded an appropriate practice from pedological viewpoint on areas with light to medium or medium-heavy texture.

It is worth emphasizing once again that while cultivation patterns of different soil types show slight variations, these variations do not reach the degree to be significant.

## Crop systems in relation to pedoclimatic conditions of the Northern Sub-Continental climate zone

Kastanozems have significantly different cropping pattern from all other soil types, which are cropped similarly to each other (Table 1, Figs. 3 and 4). High share of maize and oil crops on Kastanozems present the difference from cropping on other soils. Kastanozems are considered to be the best soils in the region, with good structure, favorable water retention and conductivity characteristics and rich nutrient reserves. Furthermore these soils situate on the southern parts of the climate zone making it favorable for crops with higher temperature needs. The areal share of maize cultivation on Kastanozems is so high, that it alone shifts the average value of corn for the entire region. As a result, the share of maize on all other soil types differs significantly. Oil crops have relatively high share on Kastanozems too. From the viewpoint of climatic suitability, rapeseed is the main oil crop to grow in this zone. Rapeseed, along with maize, is among the most demanding crops regarding nutrient uptake, especially P, of which high stock is available in Kastanozems. Cereals and root crops, on the other hand can be cultivated successfully on other soils too. Although Kastanozems would be their most suitable growing medium, higher economic return of oil crops and maize suppress cereal and root crop areas in this zone.

## Crop systems in relation to pedoclimatic conditions of the Mediterranean Semi-Arid climate zone

Great variability of crop distribution by soil types is found in the Mediterranean Semi-Arid zone (Figs. 3 and 4). Especially Acrisols, Calcisols, and Fluvisols show divergence from other soil types in their cropping patterns. Acrisols, soils with rather poor fertility and low pH are not preferred for cereal cultivation but have significantly higher share of oil crops than any other soil types. Sunflower is the predominant oil crop in this climate zone. Sunflower is tolerant to lower pH, which makes Acrisols acceptable edaphic environment for sunflower especially in contrast to maize, but also to cereals in general, which prefer soils with rather neutral reaction. Calcisols, on the contrary, have a significantly lower share

of oil crop (sunflower) and significantly higher share of cereals than the average share of Acrisols, Vertisols and Regosols. The cropping pattern of Fluvisols is significantly different from that of Acrisols and Vertisols too and these differences are caused by the low share of oil crops on Fluvisols. Both Calcisols and Fluvosols have higher shares of cereals, which is significantly higher on Calcisols, than those of all other soil types. Both Fluvisols and Calcisols are among the fertile soils of the region, although the fertility of Calcisols may be limited by the availability of trace elements, especially Fe and Zn. Nevertheless our analysis shows that the distribution of crops follow the pedoclimatic conditions in this climatic zone.

### Crop systems in relation to pedoclimatic conditions of the Southern subcontinental climate zone

There are two types of soils of which the cropping pattern deviates from the typical pattern of the zone, namely Gleysols and Podzols (Table 1, Figs. 3 and 4). Distribution of crops on Gleysoils significantly differs from that on Podzols, Histosols and Leptosols and the share of its maize growing area is significantly higher than that of any other soil type of the zone. This is explained by the hydromorphic features and consequent water regime of Gleysols. Cultivating maize on Gleysols can be successful under this climate, because water supply on Gleysols can be secured from groundwater also during the critical periods in July and August, when climatic drought is frequent and water demand of maize is the highest. Favoring maize to cereals on Gleysols have another pedoclimatic reason too. Gleysols are among the soils which are most prone to excess water, especially in early spring, which presents high risk in the cultivation. This risk can be lessened if spring crops with sowing time after the wettest period are cultivated. Majority of cereals are autumn plants under this climate, thus maize is an excellent alternative for that reason too. As the sowing time of maize is normally after the period of highest inland water risk, cultivating maize on Gleysosls can be regarded as a win-win situation.

Podzols are situated in those parts of the Southern subcontinental climate zone, which has relatively higher precipitation and consequent lower mean temperature. Probably this is one of the reasons for their unique cropping pattern including high shares of cereals, which is different from those of all other soil types in the climate zone, rather than their pedological properties. Apart from the relatively high rate of cereals, root crops (predominantly potato) which are also abundant in this zone have wide climatic suitability as well. However, in the case of root crops edaphic suitability plays an equally important role too, as the loose topsoil structure of Podzols is favorable for root crop.

### Crop systems in relation to pedoclimatic conditions of the Mediterranean (temperate and sub-oceanic) climate zone

Calcisols and Podzols are the two soil types of which the cropping pattern differs from those of most other soils in the Mediterranean (temperate and sub-oceanic) climate zone (Table 1, Figs. 3 and 4). In fact, these are the two soils with the largest (Calcisols) and smallest (Podzols) area cover of agricultural land in this climate zone. Podzols are cultivated for oil crops in significantly higher share than any other soil type under this climate, and the areal share of root crops is also rather high on Podzols. Legumes, maize and cereals, on the other

hand occupy relatively smaller areas, although not significantly smaller, than on other soils in the zone. We believe that this cultivation pattern is reflecting the suitability of Podzols for crops which require loose soil structure and have tolerance to moderately acidic pH. Calcisols, being one of the most fertile soils in the zone is mainly cultivated for cereals and oil crops, which are also the two most abundant crop groups of the zone. Although the relative share of cereals is higher and oil crops is lower than the average of the zone, these differences are not significant. The same applies for the relatively larger areas of legumes. Maize, on the other hand is cultivated on significantly smaller shares of Calcisols, than of the average of the zone. This finding suggests, that cereals and oil crops are the main plants in the rotation, with legumes and root crops playing a smaller role, just like maize, which is less considered in this soil than on others.

## Crop systems in relation to pedoclimatic conditions of the Temperate mountainous climate zone

Gleysoils are significantly different from all other soil types, due to their lower share of oil crops area and higher share of root crops (Table 1, Figs. 3 and 4). One explanation for these findings might be related to the soil geographical and genetic origin of Gleysols. In mountainous areas Gleysols are mostly located on plots with flat topography, where underground water causes reducing conditions. Cultivated Fluvisols of river valleys in this zone can have similar conditions, apart from the constant groundwater influence resulting gleyic properties. While Fluvisols show similar cropping pattern to that characteristic for the whole of the climate zone, Gleysols are used significantly less for oil crop production and more for root crops. We belive that this is due to two reasons. On the one hand, Gleysols are not very suitable for rapeseed (the most common oil crop in this climate zone), particularly if the reductive layer is at shallow to medium depth, because rapeseed needs rather deep rooting zone free of hydromorphism. Rapeseed requires good, fertile soil with high or medium pH values and it doesn't tolerate compacted soils. These two phenomena are characteristic for the Gleysols in the mountainous regions of Europe. Furthermore long winter and excessive snow cover, which are frequent in this region, are not desirable. Because of the unsteady level of the yield, the successful cultivation of this crop is not assured.

On the other hand potato (the most common root crop in this climate zone) finds suitable compartments on Gleysols where gleyic properties are below the top soil layer. Potato appreciates cool temperature and balanced climate and tolerates soil acidity too. However, this crop demands good soil management, for which the technology and traditions are available in this region. Regarding climatic conditions sugar beet production in general can be successful too, as water requirements of sugar beet can be satisfied (*Supit et al., 2010*) and Gleysoils might be suitable to grow the beet after ameliorative soil management, including drainage (as the most important action to reduce the influence of water) loosing, liming and good seedbed preparation.

One should also always keep in mind that agricultural land is rather rare in this climate zone of mountainous land. Therefore local cropping practices—which traditionally are geared towards satisfying local consumption mainly of potato—can diverge the overall

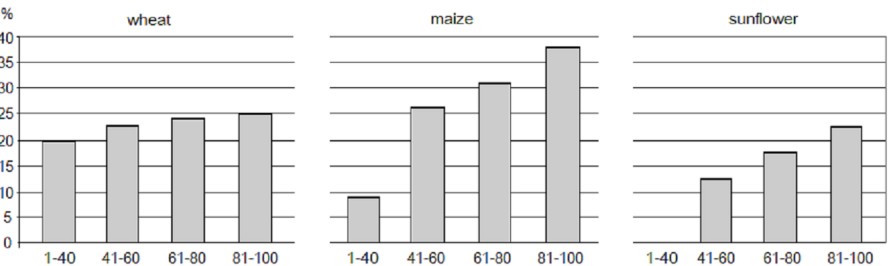

**Figure 5** Relative distribution of major crops by crop-specific soil productivity classes in Hungarian croplands.

picture to a great degree. Rootcrops on Gleysosls are concentrated in the Northern Alps and the central Carpathians. Over-presentation of root crops in this climate zone (Fig. 3) is due to the wide-ranging potato cultivation on these areas with relatively higher share of agricultural land, compared to other areas in this climate zone.

### Crop systems in relation to production suitability in Hungary

Sowing areas of all three examined crops increase by the improving soil quality (Fig. 5). Proportion of wheatland increases from 20 to 25% on areas from low to very high wheat productivity. The increase of relative sowing area of maize on higher quality soils is even larger than that of wheat. While less than 10% of low maize-productivity soils are actually used for maize cultivation, this share is above 35% on soils which are very highly productive and most suitable for maize. Although sunflower has the smallest sowing area among the three major crops in Hungary, the relative increase of sunflower land on soils from medium to very high suitability are the largest among them.

## DISCUSSION

Pedoclimatic conditions are considered in their complexity by farmers during their cropping practices across Europe. For instance oilcrops are cultivated on relatively high share of Podzols in Mediterranean (temperate-sub oceanic) and low share of Podzols in southern sub-continantal zone, meaning that similar specific soil conditions are considered together with the prevailing climatic conditions. This fact underlines the adequacy of the approach to study cropping systems by pedoclimatic conditions. Other good examples of soil-based farming include rootcrop production on Histosols in the Atlantic climate zone, maize production on Gleysosls of the Southern sub-continental climate, cultivating cereals on Podzols of the Sub-Oceanic climate zone, which all can be regarded as a "farming by soil" practice, which can be documented also on this coarse scale of analysis.

The fact that both zonal and azonal soils are among the soil types that might be cropped differently from the main cropping pattern of the given regions show that both climatic factors and soil conditions have important roles in selecting the most suitable crop. As *Sicat, Carranza & Nidumolu (2005)* found in a study area in India, farmers' perception on land suitability is largely based on their knowledge on texture, colour and rooting depth, attributes which also serve as classification criteria in soil taxonomy. Nevertheless, we have

strong reasons to believe that in Europe soil suitability-based cropping is not practiced to its full potential over the continent at the moment. For example our finding suggests that production area of legumes are not fully adapted for the local pedoclimatic conditions in some zones (eg. Temperate mountainous, Southern Sub-continental). We assume that the reason for this is not always the balanced placement of legumes with regular return after long periods to the crop rotation, but because legumes are considered mostly "only" as an interim crop between the preferred ones. Legume crops have positive rotational effects that need to be evaluated at rotational level. The reduction in the use of mineral N fertilizers in legume-supported rotations due to biological N2-fixation is the main resource benefit, which, in addition reduces greenhouse gas emissions too. Pea and Faba beans for example are relevant alternatives to soybeans in the European cropping systems and livestock diets, since they can be grown across Europe in the different pedoclimatic zones. Probably including legumes to the rotations based on pedoclimatic conditions would enhance the overall agronomical output too. However, cropping which is desirable from agronomic viewpoint do not necessarily meet the profitability targets of the farm enterprises. In order to utilize the positive agronomic and environmental benefits, the remaining gross margin deficit of legumes should be compensated or further improved e.g., with the development of new value chains and markets, improvements in agronomy and breeding. Nevertheless, the agronomic and economic performance of legumes can only be adequately evaluated when all rotational effects are taken into account, which requires the analysis of time series data, which was not available for our study.

Findings of farming in pedocimatic zones under the Atlantic climate underlines that economic drivers are decisive when farmers adopt their cropping (eg. oil crops on Albeluvsiols), however soil suitability is considered too and may result in win-win situations for the economic return of crop production and management based on soil suitability (root crops on Histosols; cereals on Arenosols).

The national scale assessment further proved that soil suitability is a major factor when farmers choose crops to cultivate. The general tendency of "higher crop specific soil productivity $\rightarrow$ higher share of the adequate crop" applies to all crops. The consciousness of farmers are reflected by the fact that wheat—a crop with the widest tolerance to climatic and soil conditions—has the least variability in its area cover among suitability classes, while the sowing areas of maize and sunflower progressively grow on soils of higher suitability. The latter two crops have more particular water and nutrient needs. Maize, a crop of high water demand grows best on deep loamy soils rich in organic matter while sunflower a crop with average water demand prefers well drained soil with easy rooting conditions (*Antal, 2005*). Although cultivation of crops on soils with different productivity show favorable distribution from the viewpoint of resource use efficiency, still, less productive land are also cultivated for the crops of our study. The lower share of cropping on low to medium quality land may have agronomic driver, namely that these major crops can be sawn in a rotation where they serve as side crops complementing the crops more suitable to grow under particular conditions. Nevertheless to reveal the full cropping pattern, including rotations needs further research.

When studying the options for optimization of cropping system for pedoclimatic conditions, we should also consider the changing climate, which influence the choice of cropping. The tendency of increased yield variability has been experienced in many regions of Europe in the last decades. Projections forecast more deteriorating agro climatic condition in terms of increased drought stress and shortening of the active growing season, which, in some regions become increasingly squeezed between a cold winter and a hot summer. Climatic changes in general are likely to shift the zonation of optimal production areas for specific crops within the EU. This tendency has implications for soil based cropping as well. Water management of soil becomes even more important aspect of farming. Soil with advantageous hydraulic properties, like Phaeozems, Chernozems and certain Luvisols and Cambisols will be the prime areas for cash crop productions. Farmers in regions of shifting climate can adopt their practices by learning the experiences from areas where suboptimal conditions has been experienced for long. In this regard the adaptability of European farmers is already demonstrated. When looking at the time series statistical data of crop cultivation (*Eurostat, 2017*) we can assume that tendencies driven by policy incentives or climate change can restructure the crop composition of pedoclimatic zones rather rapidly. To extend our study towards consideration of climate change impact and for crop-specific analysis with regional to local relevance, an analysis based on detailed agroclimatic and agropedogenic categories is recommended. As the applicability of data used in our study is limited to continental scale assessment and for one country only, such a future study shall consider more detailed climate and soil maps, both regarding spatial and thematic resolution. Improvement of thematic content of the continental data in the future by including crop-specific productivity maps (like done in the country analysis) will help to overcome the main limitation of the current continental scale study too, thus can enhance the current results for Europe which are based on pedological reasoning.

## CONCLUSIONS

Consideration of ecological conditions, including soils is key to the success and sustainability of farming. Our analysis highlights the main features of farming by soil in European pedoclimatic zones and in Hungary. Results suggest, that farmers in general, consciously take pedoclimatic condition of farming into account when selecting their cropping patterns. In other words, pedoclimatic conditions of cropping are respected and farming by soil is a common practice in all the different climatic regions of Europe. Our findings reveal, that within distinct climatic zones, soil conditions are decisive for selecting crops, and this selection follows the spatial pattern of soil distribution (Fig. 6). On the other hand, while land users need to optimize their cropping systems for the prevailing ecological conditions, economic motivations may alter the cropping practice. Our study also highlights that, this actually happens, and while European farmers crop according to the pedoclimatic conditions of their farms, economic considerations may override the ecological consideration of farming.

Future direction in the greening of the Common Agricultural Policy should include incentives that promote the optimization of soil resources use for the most profitable option that consider the local pedoclimatic conditions as well.

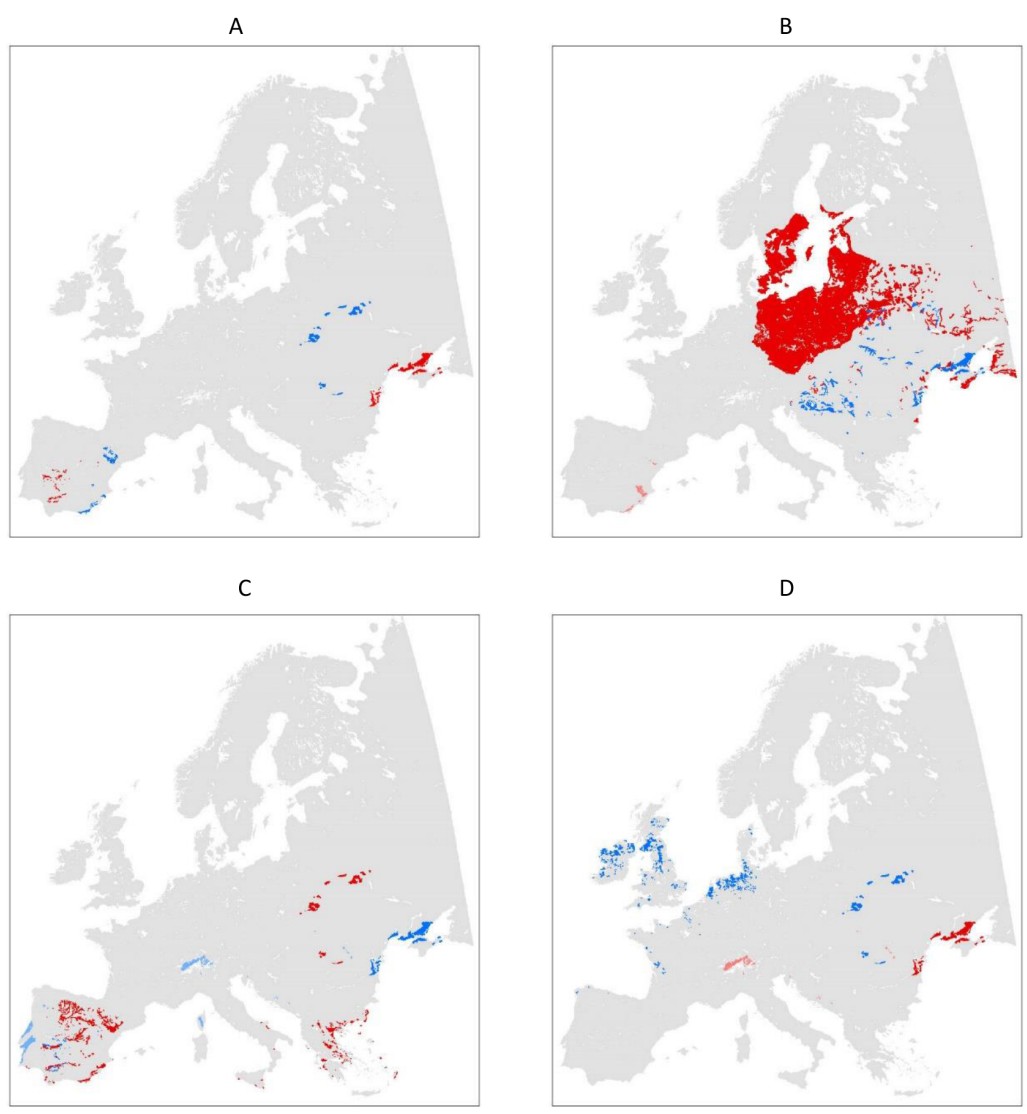

**Figure 6** **Areas of specific cropping patterns in pedoclimatic regions of Europe.** (A) Cereals; (B) maize; (C) oil crops; (D) root crops. Blue: soils with significantly larger sawing areas compared to the common areal share of this crops in the climate zone. Red: soils with significantly smaller sawing areas compared to the common areal share of this crops in the climate zone. Light blue: soils with significantly larger sawing areas compared to those of only some of the other soils in the climate zone. Rose: soils with significantly smaller sawing areas compared to those of only some of the other soils in the climate zone.

### Funding

This project was supported from the Higher Educational Institutional Excellence Program 2019 the grant of the Hungarian Ministry for Innovation and Technology (Grant Number: NKFIH-1158-6/2019) and received funding from the European Union's Horizon 2020 research and innovation programme under grant agreements: - No. 774234:

Development of Integrated Web-Based Land Decision Support System Aiming Towards the Implementation of Policies for Agriculture and Environment - No. 635750: Interactive Soil Quality Assessment in Europe and China for Agricultural Productivity and Environmental Resilience - No. 67744: Testing and promoting adaptation of soil-improving cropping systems across Europe - No. 818346: Sino-EU Soil Observatory for Intelligent Land Use Management. The funders had no role in study design, data collection and analysis, decision to publish, or preparation of the manuscript.

## Grant Disclosures

The following grant information was disclosed by the authors:
Hungarian Ministry for Innovation and Technology:  NKFIH-1158-6/2019.
Development of Integrated Web-Based Land Decision Support System Aiming Towards the Implementation of Policies for Agriculture and Environment: 774234.
Interactive Soil Quality Assessment in Europe and China for Agricultural Productivity and Environmental Resilience:  635750.
Interactive Soil Quality Assessment in Europe and China for Agricultural Productivity and Environmental Resilience: 67744.
Testing and promoting adaptation of soil-improving cropping systems across Europe: 818346.
Sino-EU Soil Observatory for Intelligent Land Use Management.

## Competing Interests

The authors declare there are no competing interests.

## Author Contributions

- Gergely Tóth conceived and designed the experiments, prepared figures and/or tables, authored or reviewed drafts of the paper, and approved the final draft.
- Tamás Kismányoky conceived and designed the experiments, authored or reviewed drafts of the paper, and approved the final draft.
- Piroska Kassai analyzed the data, prepared figures and/or tables, and approved the final draft.
- Tamás Hermann performed the experiments, analyzed the data, authored or reviewed drafts of the paper, and approved the final draft.
- Oihane Fernandez-Ugalde performed the experiments, analyzed the data, prepared figures and/or tables, authored or reviewed drafts of the paper, and approved the final draft.
- Brigitta Szabó performed the experiments, analyzed the data, prepared figures and/or tables, and approved the final draft.

## Data Availability

 The Reference Soil Groups (RSGs, *FAO, ISSS & ISRIC, 1998*) data is available at the European Soil Database (https://esdac.jrc.ec.europa.eu/resource-type/european-soil-

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
