# Peer review of "Farming by soil in Europe: status and outlook of cropping systems under different pedoclimatic conditions"

_PeerJ, doi:10.7717/peerj.8984_

## Round 0.1 · original submission · Major Revisions

The paper draws different comments from reviewers. My particular concern is from Reviewer 3, the paper needs to clarify how the soil and crops data can be used to draw conclusions.

·

Basic reporting

The article is clear, unambiguous, professional English language used throughout.

References contain 18 sources, which appropriately cover the study area.

The paper structure conforms to PeerJ standards and meets the professional article format, such as an original research, extensive data, fundamental methods (conventional and innovative) and techniques, figures, tables. Initial raw data is available.

The paper is complete and include all sections that are needed to report the key findings of the statistical analysis and validation methods referring to figures and maps.

Experimental design

Original primary research within Aims and Scope of the journal.

Research questions well defined, relevant and meaningful. Introduction discusses published data on the relationship between soil quality and present day production systems in the EU, modern concepts of sustainable development, predicted climate change affecting future agriculture. The observation allowed clearly formulate the research gap being investigated.

The submission stems on a considerable array of available data and impressive use of traditional and innovative methods, e.g. dissimilarity analysis applying Curtis dissimilarity index with “vegan” R package, “vcd” R package for analyzing the degree of association, etc. Conclusions are scientifically sound and meaningful.

Materials and method section show how the study was carried out providing details on available climate and soil databases, crops diversity and spatial distribution, statistical, GIS methods to study spatial characteristics of crop systems in relation to pedoclimatic conditions.

Validity of the findings

It is the first reserach illustrating that farmers take pedoclimatic condition of farming into account when selecting their cropping patterns in all the different climatic regions of Europe. However, economic considerations may override the ecological concern of farming. The principle impact and novelty of the study is recomendation that farmers should look at existing pedoclimatic conditions to adopt their prictice to changing climate.

The submission provides all underlying data which are available in an acceptable scientifically recognized repositories.

The conclusions summarize the results which are connected to the question investigated and supported by a well-controlled experiments.

Additional comments

The authors investigated climate categories and reference soil groups that are of a general nature and, strictly speaking, are not directly related to the assessment of suitability for agricultural activities. Perhaps it will be more informative to use agroclimatic and agropedogenic categories?

Specific comments. 1. Line 50. Authors may wish considering plural “qualities” instead of singular “quality”. 2. Lines 221,222,273. Consider plural “RSGs” instead of singular “RSG”. 3. Line 410. May be better “programmer” instead of “programme”.

Reviewer 2 ·

Basic reporting

The manuscript deals with the analysis of the crops cultivated along different soils types in different climatic zones in Europe. The topic is innovative and pertinent and the results interesting and will certainly capture attention of different background scientists who are studying agricultural activities in Europe. The text is also overall well written and organized, the structure is well defined and the Conclusions are well supported. There are, however some weaknesses that I am sure that can be easily overcome by authors in order to improve the clarity of this manuscript message:
- The number of references for this text is very low, considering especially the section Results&Discussion and Conclusion. Authors make some very nice reflections on the possible reason of how certain types of soils are more suitable to grow specific crops but often these reflections and not supported by references. This is important to show that these results have some scientific basis.
- In the Results & Discussion section, it is ok to explain the results by climate zone but, there is no need to explain in the text with bullet points the statistical differences of the soils. Instead, authors can compile the information in a nice table (or even graphics) and then briefly refer in the text, which are the main significances and important links. Otherwise, this section becomes too long and not focus.
- Conclusion section is also too long, needs to be summarized and the main findings clearly highlighted.

Experimental design

No comment

Validity of the findings

No comment

Additional comments

The manuscript deals with the analysis of the crops cultivated along different soils types in different climatic zones in Europe. The topic is innovative and pertinent and the results interesting and will certainly capture attention of different background scientists who are studying agricultural activities in Europe. The text is also overall well written and organized, the structure is well defined and the Conclusions are well supported. There are, however some weaknesses that I am sure that can be easily overcome by authors in order to improve the clarity of this manuscript message:
- The number of references for this text is very low, considering especially the section Results&Discussion and Conclusion. Authors make some very nice reflections on the possible reason of how certain types of soils are more suitable to grow specific crops but often these reflections and not supported by references. This is important to show that these results have some scientific basis.
- In the Results & Discussion section, it is ok to explain the results by climate zone but, there is no need to explain in the text with bullet points the statistical differences of the soils. Instead, authors can compile the information in a nice table (or even graphics) and then briefly refer in the text, which are the main significances and important links. Otherwise, this section becomes too long and not focus.
- Conclusion section is also too long, needs to be summarized and the main findings clearly highlighted.

Reviewer 3 ·

Basic reporting

no comments

Experimental design

The study “Farming by soil in Europe: status and outlook of cropping systems under different pedoclimatic conditions” explores the influence of soil type on the distribution of crop types in Europe. It is of scientific and general interest to assess if soil resources in Europe are used sustainable and effectively. Due to the common marked of the European Union and (at the moment) low transportation costs crops should be produced in regions in Europe that are most suitable, allowing their production with lowest input but highest yield. Due to limited quality of geo data on soils and crops the manuscript is too speculative to deliver new insights into the complex interaction of cropping paper and soil. Moreover, other influencing factors that shape the regional distribution of crops are completely ignored.

1.) An overview map (1:5 Mio; BGR [Bundesanstalt für Geowissenschaften und Rohstoffe] (2005). Soil Regions Map of the European Union and Adjacent Countries) on soil types was used as input source to describe the soils in Europe. The authors of this map write that the map presents areas with similar soil development (pedogenesis) conditions. Soil types are only rough indicators for soil quality for agricultural soil production. The relationship is weak. Within one soil types very different production potentials can be found, e.g. a Cambisol can be very sandy with low nutrient and water retention but also loamy Cambisols are found with much higher water and nutrient retention. But also different types can have similar production potentials. Moreover, the effect of cropping that affects soils (e.g. by erosion) is not accounted for in this data set. Thus, the soil type data used are not suitable to explore the cropping potential in relation to crop types.

2.) The data on the distribution of crops are derived from the MapSPAM project (You et al 2014). These geodata are based on limited data on crop types (405 units with data in Europe) and the map is described as “plausible estimation of crop distribution”. Mainly national data from FAO are used and disaggregated. Moreover, only four different cropping types classes are used to describe the croplands of Europe that expand from the Mediterranean to the Boreal. Rye, for example, requires different soils compared to wheat – however, both are combined in the class cereals. The other crop type classes get even more heterogeneous. Thus, it is questionable if this is an appropriate data source to explore the distribution of crop types in Europe. I acknowledge that this study is on continental scale, but the diversity of croplands is not captured sufficiently to explore its relation to soil properties.
3.) The paper uses a statistical approach to explore the ration between soil types and crop type. This is nicely presented in the results and discussion section together with discussion on the reasons for the anomalies that popped up. Beside soil and climate also other reasons need to be taken into account in order to reduce the speculations, e.g. animal and dairy production that is not evenly distributed in Europe but concentrates in certain regions. Food production, e.g. with maize, is thus large independent of soil and climate. Also other influencing factors are considered not systematically but anecdotally (e.g. potato production in the mountains).

Validity of the findings

The questionable data quality of the geo data (point 1 and 2) together with an limited set of drivers that influence cropping pattern (point 3) makes it hard to follow the main conclusion “we have strong reasons to believe that soil suitability-based cropping is not practiced to its full potential”. The paper need to underline this conclusion with data. This would require more transparency on uncertainties related to the data sources.

Annotated reviews are not available for download in order to protect the identity of reviewers who chose to remain anonymous.

Reviewer 4 ·

Basic reporting

The article is written in acceptable english, however different sections differ in style and grammar, indicating that sections were written by several authors.
In some sections there are large numbers of mistakes, and the spelling of.e.g. oil classification is highly erraneous in some parts.
The raw data were not available/there was no accession number provided.
Quality of references and of figures is ok, text in lines 127-138 should be added to the legends.

Specific comments:
l. 34 omit are
l. 38 areas
l. 39 underline
l. 111 the darker… the larger…
l. 112 ff: where is this citation from, and are the articles cited therein part of the original work? Is there a way to express the contect in your own words or is it necessary to cite the original? Please be specific about this.
127-138 If I understand correctly, this should be part of the legend to the respective figure(s)?
l. 150 Arensools?
l. 208 are considered
l. 208 Kastanezoms?
l. 218 Kasetnozems?
l. 234 Regoslos?
l. 252 Podzolz?
l. 292 suggests
l. 352 roles
l. 358 internal? Do you mean interim?
l. 366 does not meet
l. 371 requires
l. 380 has been
l. 396 omit are
l. 399 – 402 avoid repetitions!

Experimental design

The research question s well defined and relevant to curent discussions.
Methods are well described.

Validity of the findings

The paper aims at describing the use of major crops across soil types in Europe. Results are affirmative to most situations, and specifically Fig. 4 is important since it indicates zones of potential for future production.
The figure could have been regionalized to better show where regions for improvement are located, and the contents of this figure could have been translated to the conclusions or a recommendation subchapter.

Additional comments

interesitng paper using statistical tools to segregate areas of problematic cropping practises from well planned management schemes.

---

## Round 0.2 · Major Revisions

Because of the late review and conflicting decisions from the reviewers in the first version (Reviewer 3). I asked an additional reviewer, an expert in soil science and agricultural policy in Europe for comments. Reviewer 5 is concerned on the use of the very broad soil classes and cropping patterns. Please address this concern via stating the limitations and assumptions of this study.

Reviewer 2 ·

Basic reporting

Authors have successfully answered all the questions/comments from the first review. I am satisfied with the ending result of the manuscript and I think it will be of interest fro the scientific community.

Experimental design

Nothing to report since from the first review I have nothing to point out.

Validity of the findings

Nothing to report since from the first review I have nothing to point out.

Additional comments

Authors have successfully answered all the questions/comments from the first review. I am satisfied with the ending result of the manuscript and I think it will be of interest fro the scientific community.

Reviewer 5 ·

Basic reporting

The level of language use is good. Following the request by one reviewer, literature has been added. The context being provided is too limited in scope ( see below).

Experimental design

This paper has more the character of a review paper. The experimental design relates to the (broad) databases being considered in relation to the issue that the authors decide to raise. Here, this reviewer ( the fifth reviewer!) must agree with the critical reviewer 3. A key phrase in the article is:'soil-suitability based cropping" and: " it is not used to its full potential". Of course, farmers take pedoclimatic conditions into account when farming. They do so on the basis of experience ( "tacit knowledge") but also increasingly on scientific knowledge that reaches way beyond soil aspects. The implicit suggestion that a suitability judgment by soil scientists would guide farming practices is debatable. In fact, it is the other way around: soil scientists explored farmer experiences to define suitabilities. They did so in the past, see the FAO Land Evaluation Framework of 1976. Now the challenge of meeting the SDG targets and indicators is the name of the game. The fact that "many influencing factors shape the regional distribution of crops"( reviewer 3) is correct to the extent that trying to match very broad cropping patterns to highly generalized soil classes is doomed to fail. How about mixed cropping, very important for biodiversity, use of grassland on soils that could be well used for crops and, above all, economic considerations that the authors mention but do not elaborate on. Some soils are classified considering nutrient status and pH. That may be relevant for Africa but not for Europe where fertilization is common practice. The overall scope of this paper was intended to cover status and outlook for cropping systems. The status , as reported, is highly debatable, as mentioned, and the outlook is hardly specified. Scientifically this paper falls short of being acceptable as it does not enrich the scholarly literature. .

Validity of the findings

See above for my comments. This reviewer does not believe ( in agreement with reviewer 3) that the findings are valid. This is due to applying very general and broad databases on the one hand and a too narrow soil perspective on the other.

Additional comments

This reviewer appreciates the fact that you use soil classifications to express relations with , in this case, cropping patterns. In many soil related studies we see now introduction of seperate soil characteristics for grid points with the result that useful information on soil types gets lost. Your problem here is that you had to use very broad crop and soil patterns, that did, in the opinion of the reviewer, not allow specific conclusions as to the relation of cropping patterns with soil types. Indeed, many other factors shape the regional distribution of cropping systems. You name the important economic one but there are others such as mixed cropping, grassland use on soils that would be suitable for cropping, maize for dairy etc. I hope you will continue with your work, applying soil classification, using soil types as "carries of information". A more detailed analysis on regional level, perhaps in Hungary, could be a valuable contribution to literature.

---

## Round 0.3 · accepted · Accept

The authors have addressed the reviewer's concern and has revised the paper accordingly. There are few typos that can be fixed during production stage.